# Leptin Modulates the Response of Brown Adipose Tissue to Negative Energy Balance: Implication of the GH/IGF-I Axis

**DOI:** 10.3390/ijms22062827

**Published:** 2021-03-11

**Authors:** Vicente Barrios, Laura M. Frago, Sandra Canelles, Santiago Guerra-Cantera, Eduardo Arilla-Ferreiro, Julie A. Chowen, Jesús Argente

**Affiliations:** 1Department of Endocrinology, Instituto de Investigación La Princesa, Hospital Infantil Universitario Niño Jesús, E-28009 Madrid, Spain; laura.frago@uam.es (L.M.F.); sandra.canelles@salud.madrid.org (S.C.); santiago.guerra@estudiante.uam.es (S.G.-C.); julieann.chowen@salud.madrid.org (J.A.C.); 2Centro de Investigación Biomédica en Red de Fisiopatología de la Obesidad y Nutrición (CIBEROBN), Instituto de Salud Carlos III, E-28029 Madrid, Spain; 3Department of Pediatrics, Faculty of Medicine, Universidad Autónoma de Madrid, E-28029 Madrid, Spain; 4Department of Biological Systems, Faculty of Medicine, Universidad de Alcalá, E-28871 Alcalá de Henares, Spain; eduardo.arilla@uah.es; 5CEI UAM + CSIC, IMDEA Food Institute, E-28049 Madrid, Spain

**Keywords:** brown adipose tissue, GH axis, IGF-I signaling, leptin, lipid metabolism, UCP-1

## Abstract

The growth hormone (GH)/insulin-like growth factor I (IGF-I) axis is involved in metabolic control. Malnutrition reduces IGF-I and modifies the thermogenic capacity of brown adipose tissue (BAT). Leptin has effects on the GH/IGF-I axis and the function of BAT, but its interaction with IGF-I and the mechanisms involved in the regulation of thermogenesis remains unknown. We studied the GH/IGF-I axis and activation of IGF-I-related signaling and metabolism related to BAT thermogenesis in chronic central leptin infused (L), pair-fed (PF), and control rats. Hypothalamic somatostatin mRNA levels were increased in PF and decreased in L, while pituitary GH mRNA was reduced in PF. Serum GH and IGF-I concentrations were decreased only in PF. In BAT, the association between suppressor of cytokine signaling 3 and the IGF-I receptor was reduced, and phosphorylation of the IGF-I receptor increased in the L group. Phosphorylation of Akt and cyclic AMP response element binding protein and glucose transporter 4 mRNA levels were increased in L and mRNA levels of uncoupling protein-1 (UCP-1) and enzymes involved in lipid anabolism reduced in PF. These results suggest that modifications in UCP-1 in BAT and changes in the GH/IGF-I axis induced by negative energy balance are dependent upon leptin levels.

## 1. Introduction

Brown adipose tissue (BAT) is specialized in heat production. Its thermogenic capacity is due to the presence of uncoupling protein-1 (UCP-1) in the inner mitochondrial membrane, which disconnects mitochondrial oxidation of fuel, mainly fatty acids and glucose, from ATP synthesis, dissipating heat [1]. Brown adipocytes have a high number of mitochondria and multilocular lipid stores that provide a rapid source of fatty acids. This adipose tissue is very active in lipid metabolism, not only in lipid catabolism for heat generation, but it also has a high capacity for de novo lipogenesis, even more than white adipose cells [2].

Leptin modulates food intake, body weight, and energy expenditure, at least in part, by modulating thermogenesis. Leptin deficient ob/ob mice have low body temperature and this alteration can be amended with leptin administration [3]. Moreover, central leptin gene therapy has been shown to reduce food intake and augment expression of UCP-1 in BAT, an indicator of energy expenditure through non-shivering thermogenesis [4]. The weight reducing effects of leptin are mediated not only though reducing food intake, but also by stimulating energy expenditure. This is demonstrated by an experiment where animals are pair-fed to that of leptin treated animals, and thus had the same energy intake but lost less weight than the leptin treated animals. This phenomenon is absent in UCP-1-deficient mice, suggesting that BAT thermogenesis is required for food intake-independent leptin induced weight loss [5].

Central leptin infusion stimulates growth hormone (GH) secretion [6], most likely through inhibition of somatostatin (SRIF) expression and secretion [7]. The SRIF system inhibits pituitary function, mostly reducing hormone secretion, although it also reduces GH mRNA content [8]. Thus, reducing SRIF would increase circulating insulin-like growth factor I (IGF-I), linking the central effect of leptin with the peripheral IGF system.

IGF-I can modify BAT thermogenesis, with the lack of the IGF-I receptor (IGF-IR) reducing UCP-1 in interscapular brown fat and leading to impaired cold acclimation [9]. Moreover, IGF-I deficiency and malnutrition, which decreases IGF-I production, are associated with reduced UCP-1 levels and defective thermogenesis [10]. Our experimental model increases peripheral leptin [11] and crosstalk between leptin and IGF-I has been reported in different tissues [12,13]. Activation of insulin/IGF-I pathways stimulates changes in fatty acid metabolism, essential during thermogenesis, as they are required for UCP-1 activity in BAT mitochondria. One mechanism to increase lipid concentrations is via lipogenesis, and activation of signaling targets of IGF-I can promote expression of key enzymes involved in this anabolic process [14].

We hypothesized that increased central leptin levels can modulate thermogenesis through modifications in the GH/IGF-I axis and changes in IGF-I-related signaling in BAT. Thus, the aims of this study were to compare the effect of chronic central leptin infusion with pair-fed rats on the GH/IGF-I axis and to characterize the changes in activation of leptin- and IGF-I-related signaling and markers of lipid metabolism and thermogenesis. The group of pair-fed rats were included in order to discriminate between the direct effects of leptin and those due only to a reduction in food intake.

## 2. Results

### 2.1. General Characteristics of Experimental Groups

We have previously reported that food intake and body weight gain were reduced in the PF and L groups [11]. Free IGF-I was undetectable and C and PF groups and interscapular BAT mass reduced in PF rats. Glycemia, insulinemia, relative mRNA levels of peroxisome proliferator-activated receptor-γ (PPARγ), and protein levels of different signaling targets, such as the beta chain of the insulin receptor (IRβ), protein tyrosine phosphatase 1B (PTP1B) and phosphorylated (p) protein tyrosine phosphatase 1B (STAT5) in BAT were not found to differ among the experimental groups (Table 1).

### 2.2. Effects of Food Deprivation and Leptin Infusion on the GH/IGF-I Axis

Hypothalamic SRIF mRNA levels were increased in the PF group and reduced by leptin treatment compared to C and PF groups (Figure 1A). Somatostatin receptor 2 (sst2) modulates the inhibitory effect of SRIF on GH synthesis [15]; thus, we measured its mRNA levels in the pituitary gland. Both pair-feeding and leptin-treated rats presented a decrease in sst2 mRNA levels (Figure 1B). Pituitary GH mRNA levels were only reduced in the PF group (Figure 1C).

Hypothalamic concentrations of SRIF were reduced in leptin-treated rats (Figure 1D) and pituitary levels of sst2 were diminished in PF and L groups (Figure 1E). Protein levels of GH in the pituitary were reduced in PF and L groups, with a greater decrease in the L group (Figure 1F). Serum GH levels were diminished in the PF group (Figure 1G).

Hepatic growth hormone receptor (GHR) protein levels were not different between the experimental groups (Figure 1H). Phosphorylation of the signal transducer and activator of transcription 5 (STAT5) was reduced in the liver of PF rats (Figure 1I). Finally, we studied serum IGF-I levels, which were diminished in PF rats (Figure 1J).

### 2.3. Effects of Central Leptin Infusion on BAT

No differences in the levels of phosphorylation of the insulin receptor (IR) were seen among the experimental groups (Figure 2A). Chronic central leptin infusion or pair feeding did not modify protein levels of the IR (Table 1).

The phosphorylation of IGF-IR on tyrosine is a key step to activating this receptor pathway. Phosphorylation of IGF-IR on Tyr1131 residue was increased in the L group (Figure 2B), as was the phosphorylation of insulin receptor substrate 1 (IRS1) on Tyr residues (Figure 2C). In addition, the phosphorylation of protein kinase b (Akt) on Thr308 (Figure 2D) and Ser473 (Figure 2E) was increased by central leptin infusion, while phosphorylation of Akt on Thr308 was reduced in PF animals.

Phosphorylation of Akt can activate cAMP response element binding protein (CREB) through phosphorylation on its Ser133 residue [16], with CREB activation being increased in leptin-treated rats (Figure 2F). The UCP-1 promoter contains a cyclic AMP response element [17] and mRNA levels of UCP-1 were decreased in PF rats, with no effects in leptin-treated rats (Figure 2G).

### 2.4. Leptin and IGF Signaling in BAT

There were no changes in the protein levels of the long form of the leptin receptor (ObRb) (Figure 3A). Phosphorylation of signal transducer and activator of transcription 3 (STAT3) on the Ser727 residue was reduced in the L group with respect to the C and PF groups (Figure 3B) and suppressor of cytokine signaling 3 (SOCS3) levels were diminished in both the PF and L groups (Figure 3C). As SOCS3 can interact with IGF-IR [18], we analyzed the association between SOCS3 and IGF-IR. Immunoprecipitation studies showed that the association between these proteins was reduced in leptin-treated rats (Figure 3D). There were no changes in mRNA levels of IGF-I (Figure 3E). However, when IGF-I concentrations in interscapular BAT were analyzed, an increase in the L group was found (Figure 3F). Finally, free IGF-I was undetectable in C and PF rats but measurable in leptin-treated rats (Table 1).

### 2.5. Central Leptin Infusion Effects on BAT Inflammatory Markers

When the activation of signaling targets related to inflammation was studied, a reduction in c-Jun N-terminal kinase (JNK) and nuclear factor kappa B (NFkB) phosphorylation in BAT of leptin-treated rats was found (Figure 4A,B, respectively). Activation of these signaling pathways can change the expression and levels of pro- and anti-inflammatory factors. Levels of IL-6 were reduced in leptin-treated rats (Figure 4C) and tumor necrosis factor-α (TNF-α) concentrations were diminished in both PF and L groups, with lower concentrations in the latter (Figure 4D). Levels of IL-4 were augmented in leptin-treated rats (Figure 4E).

### 2.6. Changes in Variables Involved in Lipid Metabolism

Relative mRNA levels of glucose transporter 4 (GLUT4) were higher in L compared to both C and PF rats (Figure 5A). The activity of malic enzyme was reduced in PF with respect to C and L groups (Figure 5B) and no changes in the mRNA levels of lipoprotein lipase (LPL) were detected (Figure 5C). Central leptin infusion has been shown to change systemic tissue levels and activity of enzymes involved in lipid metabolism [19]. Here, phosphorylation of ATP-citrate lyase (ACL) at Ser 455 was increased in L rats with respect to the PF group (Figure 5D) and relative mRNA levels of acetyl CoA carboxylase-α (ACCα) were decreased in the PF with respect to the C and L groups (Figure 5E). Activation of ACC, measured as phosphorylation at Ser79 residue, showed no differences (Figure 5F). mRNA levels and relative protein content of fatty acid synthase (FASN) were decreased in the PF group (Figure 5G,H, respectively). Lastly, no differences in carnitine palmitoyl transferase 1b (CPT1b) mRNA levels were observed among the experimental groups (Figure 5I).

## 3. Discussion

Our results show that central leptin infusion results in modifications in the GH/IGF-I axis and this is associated with modifications in markers of inflammation and metabolism in BAT that are not observed in pair-fed rats, suggesting a specific effect of leptin that is not associated with the reduction in food intake. The changes in the activation of IGF-I-related intracellular targets in BAT are associated with a reduction in UCP-1 expression induced by a negative energy balance. Finally, our results suggest a possible relationship between the reduction in inflammatory factors and the activation of IGF-I signaling in leptin-treated rats that may be associated with an increase in the expression and activity of enzymes implicated in fatty acid synthesis in BAT.

Central infusion of leptin did not affect circulating GH levels in a situation of caloric restriction, which contrasts with that observed in food-restricted animals. In response to chronic leptin infusion, pituitary GH expression was unchanged, although there was a reduction in the mRNA levels of hypothalamic SRIF and pituitary sst2, a SRIF subtype receptor that mediates inhibitory actions of this neuropeptide on GH synthesis [20]. The differences between pair-fed and leptin-treated rats in serum GH levels could be due not only to the reduced SRIF tone in leptin-treated rats, but also to the possible contribution of GH-releasing hormone, as central leptin infusion augments hypothalamic GH-releasing hormone in a similar manner as GH levels [21]. Serum GH levels tend to be higher in leptin-treated rats and this upward tendency may be due to a greater level of GH secretion, as suggested by the reduced GH protein concentration in the pituitary of leptin-treated rats with respect to controls. These data are in agreement with the diminished SRIF mRNA levels in the leptin group, since SRIF inhibits not only the synthesis, but also the secretion of GH [22].

Although both groups have a reduction in food intake, the leptin-infused group has an excess of hypothalamic leptin, which would indicate a positive energy balance to the neuronal circuits controlling metabolic homeostasis. In contrast, the pair-fed rats have reduced leptin levels [22], with this difference possibly explaining some of the differences observed between these two experimental groups. We found a reduction in STAT5 activation in the liver of pair-fed rats, parallel to the pattern of GH levels, which is in accordance with the reduction of serum IGF-I found in these rats. In fact, STAT5 is a key target in GH signaling involved in the synthesis of IGF-I [23].

Evidence suggests that IGF-I induces UCP-1 expression and thermogenesis, whereas treatment with an inhibitor of an IGF-IR tyrosine kinase blocks this effect [24]. In leptin-treated rats, we found an increase in the phosphorylation of IGF-IR on Tyr 1131, a key residue as mutation resulting in deficient phosphorylation of residue Tyr1131 in this receptor abrogates its activation [25]. The activation of IRS-1 in leptin-infused animals seem to be relevant for subsequent activation of this pathway, which could affect thermogenesis in BAT. Indeed, IRS-1^−/−^ brown adipocytes are unable to induce Akt phosphorylation and expression of UCP-1 [26].

Although phosphorylation of Akt on Thr308 has been classically attributed to insulin, IGF-I is also implicated in this step of activation of this intracellular pathway [27]. Moreover, here we have found no changes in either phosphorylation or total protein levels of insulin receptor. Activation of Akt needs the phosphorylation of Thr308 in the activation loop by the phosphoinositide-dependent kinase 1 and in Ser473 residue, as we have detected in leptin-treated rats, whereas pair-fed group presented lower activation in Thr308 residue. There is evidence that activation of the Akt pathway leads to CREB phosphorylation through different stimuli [28], and importantly, CREB has been identified as a regulatory element in the UCP-1 promoter [29]. Thus, our results indicate that although leptin infused rats had a chronic reduction in food intake, the high leptin preserved the expression of UCP-1 in BAT, which seem be related with changes in the Akt pathway, while in the pair-fed rats the negative energy balance and the physiological reduction in leptin levels results in decreased UCP-1 in this tissue. In addition, we have previously found an increase in circulating ghrelin in pair-fed rats [11] that could reduce the UCP-1 function, as ghrelin administrated intravenously suppresses noradrenaline release in BAT, whereas genetic suppression of ghrelin receptors activates the BAT function [30].

Attenuation of leptin signaling in BAT could be involved in the activation of the IGF-I pathway. Deletion of SOCS3 increases phosphorylation of IRS1 and Akt in other tissues [31] and central overexpression of leptin reduces STAT-3 activation in male rats [32]. We found a reduction in STAT-3 phosphorylation and SOCS-3, as well as a fall in the association of SOCS3 to IGF-IR in leptin-treated rats that may be explained, at least in part, by the increased phosphorylation of IGF-IR and the subsequent activation of this pathway. The changes in IGF-I-related signaling may be related with circulating IGF-I concentrations, as we did not find variations in the mRNA levels of IGF-I in BAT. It is interesting to note that central infusion of leptin augments serum IGF-I levels in ob/ob mice [33] and an increase in the concentration of free IGF-I at the cell surface allows the activation of IGF-I receptor [34].

Changes in the inflammatory environment in BAT might be expected to modify insulin/IGF-I signaling and to alter thermogenesis in BAT. In this regard, TNFα inhibits tyrosine phosphorylation of IRS-1 and increases it on serine residues, impairing insulin/IGF-I action [35]. Inflammation causes insulin resistance in brown adipocytes and suppresses the induction of UCP-1 gene expression in BAT [36]. Moreover, genetic ablation of JNK-1, another major intracellular mediator of inflammatory signaling, enhances UCP-1 expression in adipose tissues [37].

In conjunction with the reduction in inflammatory markers in leptin-treated rats, we found an increase in IL-4, an anti-inflammatory interleukin. Nguyen et al. [38] demonstrated a requirement for IL-4-stimulated macrophage activation in adaptive thermogenesis. Exposure to cold was shown to promote the secretion of catecholamines by macrophages, favoring thermogenic gene expression in BAT. The lack of these activated macrophages impaired metabolic adaptation to cold, whereas exogenous IL-4 augmented the expression of thermogenic markers. In addition, a rise in IL-4 after leptin treatment may be an additional mechanism by which Akt is activated, as IRS1 is phosphorylated by this interleukin in adipocytes [39]. Thus, together these data suggest that the reduction in inflammatory markers and their signaling in leptin-treated rats may link the activation of IGF-I signaling and thermogenesis.

Activity of BAT is reduced in obesity and diabetes, indicating that this tissue participates in diet-induced thermogenesis [40]. In a situation of caloric restriction, fatty acid synthesis is diminished; thus, the source of the substrate for mitochondrial electron transport chain and generation of heat is reduced. Our results show that the activation of ACL, as well as the expression of ACC and FASN, key enzymes of lipogenesis are reduced in pair fed rats but not in leptin-treated rats. However, we found no changes in the expression of LPL, suggesting that leptin infusion increases the levels of fatty acids in BAT due to higher biosynthesis and not due to LPL contribution in these adipose cells. Previous findings demonstrate that leptin deficiency in ob/ob mice results in reduced energy expenditure achieved through reduced BAT function. These mice present diminished thermogenic markers associated with lower expression of the main enzymes implicated in fatty acid synthesis as well as those related to lipid oxidation [41].

Whereas leptin deficiency reduces insulin/IGF-I signaling, a decrease in leptin-related signaling may favor central and peripheral insulin sensitivity [31,42]. In BAT, we have demonstrated a reduction in leptin signaling concomitant to the lower association of SOCS3 to IGF-IR. Improvement of insulin/IGF-I signaling is reported to stimulate lipogenesis in BAT and Akt is crucial for promoting maintenance and adipogenesis in BAT [43]. In addition to using lipids, BAT has a high rate of glucose uptake to be used as fuel or lipid synthesis. Chronic central leptin infusion increases insulin-related glucose metabolism and augments its uptake [44]. We found an increase in the expression of GLUT4 in leptin-treated rats and IGF-I has been shown to augment its expression in brown adipocytes [45]. Thus, this possible increase in glucose uptake could be used, at least in part, for the generation of NADPH, necessary for lipid synthesis. Our results show no change in the activity of malic enzyme in leptin-treated rats, whereas pair feeding reduces this enzymatic activity. In this regard, it has been reported that increased serum leptin levels augment malic enzyme activity [46] and our experimental model of leptin treatment has higher serum leptin concentrations that could possibly induce this enzyme [11].

Whether the GH/IGF-I axis is directly responsible for the changes related to thermogenesis deserves further attention. One must take into consideration that in these studies we examined not only the changes in IGF-I-related signaling, but also the leptin-related signaling in BAT. Additional studies are needed to establish the direct contribution of each factor to the modulation of thermogenesis by leptin.

In a situation of negative energy balance, the physiological response is to reduce energy expenditure. Here we show that BAT, an important tissue in the regulation of energy expenditure, responds differently to energy deficit depending on leptin levels.

## 4. Materials and Methods

### 4.1. Materials

All chemicals were purchased from Merck (Darmstadt, Germany) unless otherwise noted. Specific antibodies against actin were from Thermo Fisher Scientific (Waltham, MA, USA), PTP1B from Merck, GHR and phosphorylated (p) ATP citrate lyase (pACL) from Cell Signaling Technology (Danvers, MA, USA), pACC, FASN, ObRb, vinculin and sst2 from Santa Cruz Biotechnology (Santa Cruz, CA, USA) and SOCS3 from Proteintech Europe (Manchester, UK). The Immun-Star Western C kit was from Bio-Rad Laboratories (Hercules, CA, USA). The corresponding secondary antibodies conjugated with horseradish-peroxidase, the high-capacity cDNA kit and TaqMan gene expression assays were purchased from Thermo Fisher Scientific.

### 4.2. Animals

All procedures were carried out in accordance with the local ethics committee and complied with Royal Decree 53/2013 pertaining to the protection of experimental animals and with the European Communities Council Directive (2010/63/EU). The study was approved by the Ethical Committee of Animal Experimentation of the Universidad de Alcalá (PROEX018/16, 14 June 2016). All care was taken to use the minimum number of animals. Male Wistar rats (250 ± 10 g) purchased from Harlan Laboratories (Barcelona, Spain) were individually caged and fed standard chow and water ad libitum. Animals were anesthetized using 4 mg of ketamine/100 g bw and 0.5 mg of xylazine/100 g bw throughout chirurgical procedures.

### 4.3. Experimental Design

After an overnight fast, 15 rats were anesthetized (0.02 mL of ketamine/100 g wt and 0.04 mL of xylazine/100 g wt) and positioned in a stereotaxic apparatus and treated, as previously reported [47]. A cannula attached to an osmotic minipump (Alzet, Durect Corp., Cupertino, CA, USA) containing either saline or leptin was implanted into the right cerebral ventricle (−0.3 mm anteroposterior, 1.1 mm lateral from Bregma). Leptin was dissolved in saline plus 1% BSA. Rats were treated *icv* for 14 days with either saline with 1% BSA or leptin (12 μg/day). As leptin treatment reduces food intake, we also included a pair-fed group that received the same amount of food consumed by the leptin-treated group the day before. This resulted in three groups: chronic saline with 1% BSA (control, C), pair-fed rats with chronic saline with 1% BSA (PF) and chronic leptin (L). Rats were sacrificed by decapitation at 8.00 h after a 12 h fasting period and the hypothalamus, pituitary, liver and interscapular BAT were isolated, weighed and processed. Trunk blood was collected and centrifuged at 1800× *g* for 10 min at 4 °C and serum was collected and stored at −70 °C.

### 4.4. Tissue Homogenization and Protein Quantification

For immunodetection of pACL, pThr308Akt, pSer473-Akt, Akt, GHR, p-IR), β-chain of the IR (IRβ), interleukin (IL)-4, IL-6, pTyr1131 insulin-like growth factor-I receptor (pTyr1131IGF-IR), pThr183/Tyr185-JNK (pThr183/Tyr185-JNK), JNK, pSer536 NFkB (pSer536NFkB), NFkB, ObRb, PTP1B, SOCS3, pTyr694- STAT5 (pTyr694STAT5), STAT5, sst2, and TNF-α, liver and interscapular BAT were homogenized on ice in 400 µL of lysis buffer (Merck). The lysates were incubated overnight at −80 °C. Samples were centrifuged at 12,000× *g* for 5 min at 4 °C and the supernatants were stored at −80 °C until assayed. Protein concentration was determined by the Bradford method (Bio-Rad Laboratories).

### 4.5. ELISAs

#### 4.5.1. Phosphorylation of Insulin Receptor

The phosphorylation of the insulin receptor was measured by an ELISA kit from Assay Solution (Woburn, MA, USA). Lysates of BAT were incubated for 2 h at RT and after washing, a detection antibody coupled to biotin was incubated in the microplate during 2 h at the same temperature. A streptavidin-HRP complex was added and finally, after washing, the samples were incubated with tetramethylbenzidine until the absorbance was read at 450 nm.

#### 4.5.2. Phosphorylation of IGF-I Receptor

The ELISA employed (Cell Signaling Technology) detects levels of IGF-I receptor β protein when phosphorylated at Tyr1131 residue. Briefly, tissue lysates were incubated for 2 h at 37 °C in a microplate coated with the pTyr1131-IGF-IRβ antibody. After washing, a detection antibody was added and incubated at 37 °C for 1 h. Then, the microplate was washed again and an HRP-linked secondary antibody was added and incubated at the same temperature for 30 min. Finally, washing was performed, the substrate was added, and the absorbance was read at 450 nm.

#### 4.5.3. Hypothalamic SRIF Concentrations

Concentrations of SRIF in the hypothalamus were determined with an ELISA kit from Cusabio (Wuhan, China). Hypothalamic lysates were incubated during 2 h at 37 °C in a microplate pre-coated with an antibody specific for SRIF. After removing any unbound substances, a biotin-conjugated antibody was added and incubated again during 1 h at 37 °C. After washing, avidin-horseradish peroxidase (HRP) was added and incubated 1 h at 37 °C. Following a wash, a substrate solution was added, and the absorbance was read at 450 nm.

#### 4.5.4. Pituitary GH and Serum GH and Insulin Levels

Serum GH and insulin concentrations were measured using ELISA kits from Merck according to the manufacturer’s instructions.

#### 4.5.5. Free and Total IGF-I in Tissue Samples

Free IGF-I was determined using an ELISA kit from AnshLabs (Webster, TX, USA). Total levels of IGF-I were measured with an ELISA kit from Mediagnost (Reutlingen, Germany). Both assays were performed according to the manufacturer’s instructions.

Intra- and inter-assay variation coefficients were lower than 10% in all assays.

### 4.6. Western Blotting

For the detection of phosphorylated ACC, FASN, GHR, ObRb, PTP1B, sst2 and SOCS3, protein was resolved on 10% SDS-polyacrylamide gels. The proteins were transferred to polyvinyl difluoride membranes and blocked with Tris-Tween buffered saline (TTBS) containing 5% (wt/vol) BSA during 2 h at 25 °C and incubated with the corresponding primary antibody (diluted 1:1000) in TTBS with BSA at 4 °C overnight. Membranes were washed and incubated with the corresponding secondary antibody conjugated with peroxidase at a dilution of 1:2000 in TTBS with BSA during 90 min at 25 °C. Proteins were detected by chemiluminiscence using an ECL system (Bio-Rad Laboratories, Hercules, CA, USA). Quantification of the bands was carried out by densitometry using a ImageQuant LAS4000 mini TL Software (GE Healthcare Bio-Sciences AB, Sweden). Gel loading variabilities for GHR, ObRb, PTP1B and SOCS3 were normalized with actin, pACC, ACL and FASN with vinculin and sst2 with tubulin.

### 4.7. Immunoprecipitation

The association between SOCS3 and IGF-IR was studied by immunoprecipitation. One hundred µg of BAT was homogenized on ice in 400 μL of lysis buffer pH 7.6 containing 50 mM Hepes, 10 mM EDTA, 50 mM sodium pyrophosphate, 100 mM NaF, 10 mM Na3VO4, 1% Triton X-100, 2 mM phenylmethylsulfonyl fluoride, 10 μg/mL leupeptin, and 10 μg/mL aprotinin. Equal amounts of protein (400 μg) of each sample were immunoprecipitated overnight at 4 °C with an IGF-IR antibody (Oncogene Research Products, La Jolla, CA, USA) and then incubated with Protein A-agarose beads for 2 h at 4 °C. Immunocomplexes were washed three times with lysis buffer, extracted for 5 min at 95 °C in 4X SDS-PAGE sample buffer (200 mM Tris-HCl, 12% SDS, 4 mM EDTA, 8% 2-mercaptoethanol, 20% glycerol, pH 7.6) and SOCS3 analyzed by Western blotting, as described above.

### 4.8. Multiplexed Bead Immunoassay

Phosphorylated and total protein levels of Akt, CREB, IRS1, JNK, and NFkB, as well as the content of IL-4, IL-6, and TNF-α in interscapular BAT and both forms of signal transducer and activator of transcription (STAT) 3 and 5 in BAT and 5 in liver were measured in duplicate by multiplexed bead immunoassays (Bio-Rad Laboratories and Merck), as previously reported [48]. Briefly, magnetic beads conjugated to the appropriate antibodies and tissue lysates (50 μL each) were incubated for 18 h at 4 °C. Afterwards, wells were washed using a magnetic separation block (Millipore) and antibody conjugated to biotin (25 μL) was added. After incubation for 30 min at room temperature, beads were incubated during 30 min with 50 μL streptavidin conjugated to phycoerythrin. A minimum of 50 beads per parameter were analyzed in the Bio-Plex suspension array system 200 (Bio-Rad Laboratories). Raw data (median fluorescence intensity, MFI) were analyzed with the Bio-Plex Manager Software 4.1 (Bio-Rad Laboratories). Mean intra- and inter-assay coefficients of variation were 8.3% and 11.9%, respectively.

### 4.9. Enzyme Activity Assay

Activity of malic enzyme [EC 1.1.1.40] was determined according the method of Geer et al. [49]. Briefly, after homogenization of 20 mg of liver in PBS and subsequent centrifugation, diluted supernatants were incubated at 25 °C with a triethanolamine buffer, malic acid, and NADP^+^, and the absorbance at 340 nm was monitored.

### 4.10. RNA Purification and Real-Time PCR Analysis

Total RNA was extracted from the hypothalamus, pituitary, and interscapular BAT according to the Tri-Reagent protocol [50]. The reverse transcription reaction was carried out on 2 μg of RNA using a high-capacity cDNA archive kit (Applied Biosystems, Foster City, CA, USA). Real-time PCR was performed in an ABI Prism 7000 Sequence Detection System (Applied Biosystems) using TaqMan PCR Master Mix and the thermocycler parameters recommended by the manufacturer. PCRs were performed in a volume of 50 μL, containing 25 μL of the reverse transcription reagents. TaqMan gene expression assays were used for ACCα, CPT1b, FASN, GH, GLUT4, IGF-IR, PPARγ, SRIF, sst2, and UCP-1 (Rn00573474_m1, Rn00682395_m1, Rn00569117_m1, Rn01495894_g1, Rn00562597_m1, Rn01477918_m1, Rn00440945_m1, Rn00561967_m1, Rn00571116_m1 and Rn00562126_m1, respectively; Applied Biosystems). Relative gene expression comparisons were carried out using an invariant endogenous control (actin, Rn00667869_m1). The ΔΔCT method was used for relative quantification.

### 4.11. Statistical Analysis

Analysis of all data was carried out by one-way ANOVA followed by Bonferroni’s post-hoc tests. Values were considered significantly different when the p value was less than 0.05. Statistical analyses were performed using Statview software (Statview 5.01, SAS Institute, Cary, NC, USA). Data are presented as mean ± SEM.

## 5. Conclusions

As summarized in Figure 6, our results suggest that leptin infusion maintains thermogenesis in a situation of caloric restriction. Changes in lipid anabolism play a key role in this process, and this could be due to activation of IGF-I signaling, which may be associated with the reduction in the association of SOCS3 to the IGF-IR and the improvement of the inflammatory environment in BAT as well as to the preservation of the GH-IGF-I axis.

## Figures and Tables

**Figure 1 ijms-22-02827-f001:**
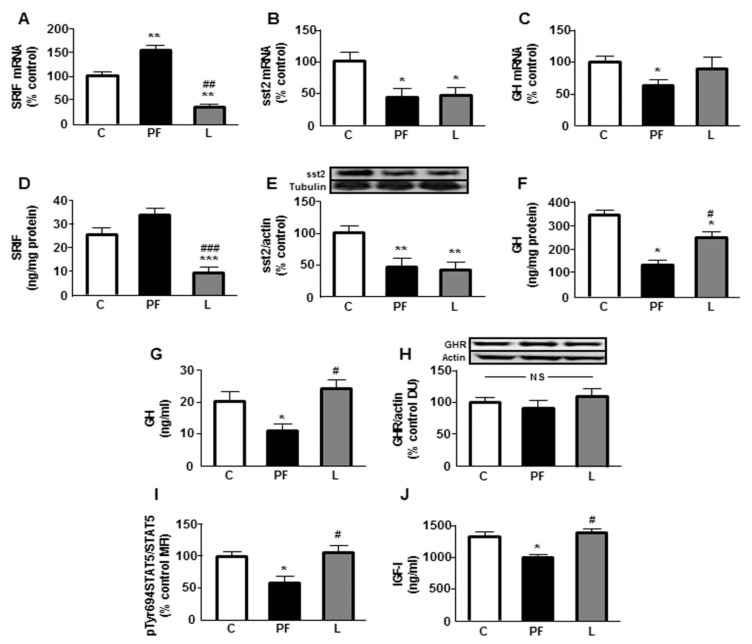
Effects of central leptin infusion on the growth hormone (GH)/insulin-like growth factor I (IGF-I) axis. (**A**) Relative hypothalamic somatostatin (SRIF) mRNA content, (**B**) pituitary somatostatin receptor subtype (sst)2 mRNA content, (**C**) pituitary GH mRNA content, (**D**) SRIF protein levels in the hypothalamus, (**E**) sst2 relative protein levels in the pituitary, (**F**) GH protein levels in the pituitary, (**G**) serum GH concentration, (**H**) GH receptor (GHR) protein levels in the liver, (**I**) protein levels of signal transducer and activator of transcription (STAT)5 phosphorylated (p) at Tyr694 (pTyr694STAT5) in the liver and (**J**) serum IGF-I concentrations in control rats (C), pair-fed rats (PF) and rats receiving a chronic icv leptin infusion (L). Data are presented as means ± SEM. N = 5. DU, densitometry units; MFI, median fluorescent intensity; NS, non-significant. * *p* < 0.05, ** *p* < 0.01, *** *p* < 0.001 vs. C, ^#^
*p* < 0.05, ^##^
*p* < 0.01, ^###^
*p* < 0.001 vs. PF.

**Figure 2 ijms-22-02827-f002:**
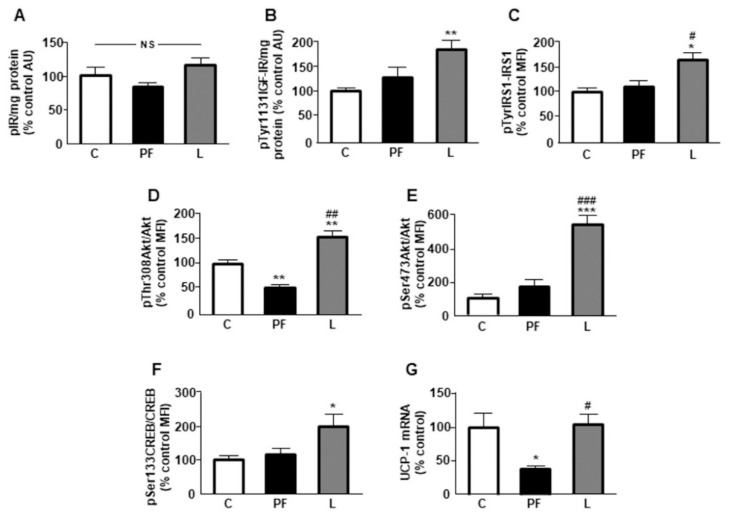
Effects of central leptin infusion on intracellular signaling in brown adipose tissue (BAT). Relative protein levels of (**A**) phosphorylated (p) insulin receptor (pIR), (**B**) insulin-like growth factor-I receptor (IGF-IR) phosphorylated at Tyr1131 (pTyr1131IGF-IR), (**C**) insulin receptor substrate 1 (IRS1) phosphorylated at Tyr residues, (**D**) Akt phosphorylated at Thr308 (pThr308Akt), (**E**) Akt phosphorylated at Ser473 (pSer473Akt), (**F**) cAMP response element binding protein (CREB) phosphorylated at Ser133 (pSer133CREB) and (**G**) relative uncoupling protein (UCP)-1 mRNA content in the interscapular BAT of control rats (C), pair-fed rats (PF) and rats receiving a chronic icv leptin infusion (L). Data are presented as means ± SEM. N = 5. AU, absorbance units, MFI, median fluorescent intensity; NS, non-significant. * *p* < 0.05, ** *p* < 0.01 *** *p* < 0.001 vs. C; ^#^
*p* < 0.05, ^##^
*p* < 0.01, ^###^
*p* < 0.001 vs. PF.

**Figure 3 ijms-22-02827-f003:**
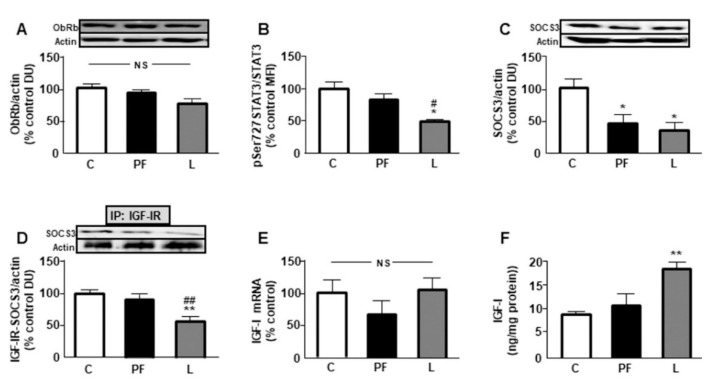
Leptin signaling and IGF-I in brown adipose tissue (BAT). (**A**) Relative protein levels of the long form of the leptin receptor (ObRb), (**B**) phosphorylated (p) signal transducer and activator of transcription 3 at Ser727 (Ser727STAT3), (**C**) suppressor of cytokine signaling-3 (SOCS3), and (**D**) SOCS3 associated to insulin-like growth factor (IGF)-I receptor (IGF-IR) and (**E**) relative IGF-I mRNA content and (**F**) IGF-I concentrations in interscapular BAT of control rats (C), pair-fed rats (PF) and rats receiving a chronic icv leptin infusion (L). Data are presented as means ± SEM. N = 5. DU, densitometry units; IP, immunoprecipitation; MFI, median fluorescent intensity; NS, non-significant. * *p* < 0.05, ** *p* < 0.01 vs. C; ^#^
*p* < 0.05, ^##^
*p* < 0.01 vs. PF.

**Figure 4 ijms-22-02827-f004:**
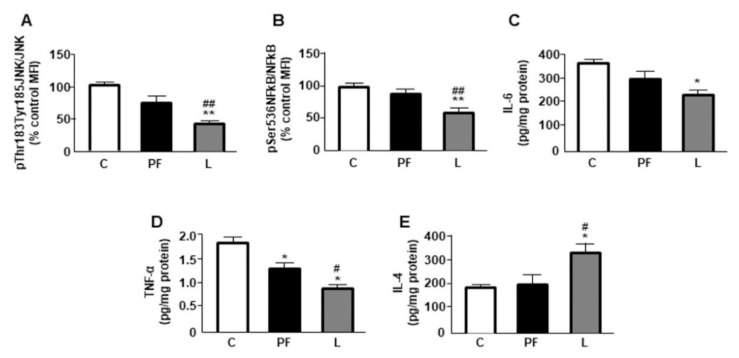
Effect of chronic central leptin treatment on inflammatory markers in brown adipose tissue (BAT). (**A**) Relative protein levels of c-Jun N-terminal kinase (JNK) phosphorylated (p) at Thr183/Tyr185 (pThr183/Tyr185JNK), and (**B**) nuclear factor kappa B (NFkB) phosphorylated at Ser536 (pSer536NFkB), and (**C**) interleukin (IL)6, (**D**) tumor necrosis factor (TNF)α and (**E**) IL-4 content in the interscapular BAT of control rats (C), pair-fed rats (PF) and rats receiving a chronic icv leptin infusion (L). Data are presented as means ± SEM. N = 5. MFI, median fluorescent intensity. * *p* < 0.05, ** *p* < 0.01 vs. C; ^#^
*p* < 0.05, ^##^
*p* < 0.01 vs. PF.

**Figure 5 ijms-22-02827-f005:**
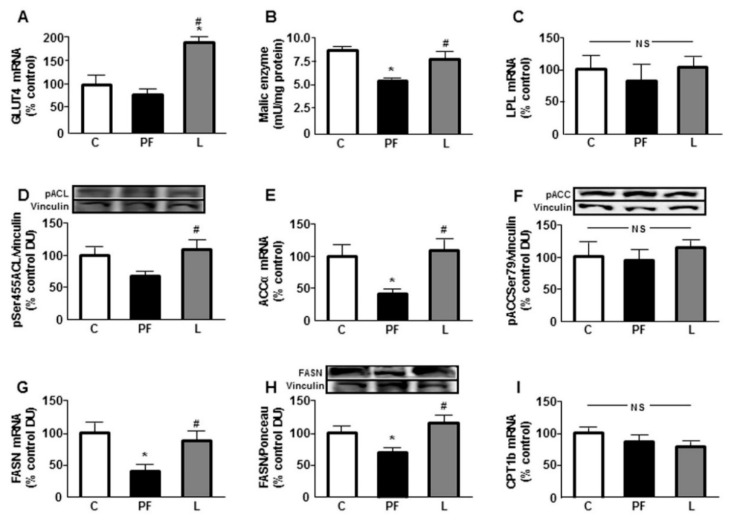
Metabolic response of brown adipose tissue (BAT) to icv leptin infusion. Relative (**A**) glucose transporter (GLUT)4 mRNA, (**B**) malic enzyme activity, (**C**) lipoprotein lipase (LPL) mRNA, (**D**) phosphorylated (p) protein levels of ATP citrate lyase (ACL) at Ser455 (pSer455ACL), (**E**) acetyl-CoA carboxylase (ACC)α mRNA, (**F**) phosphorylated protein levels of ACC at Ser79 (pACCSer79), (**G**) fatty acid synthase (FASN) mRNA, (**H**) protein levels of FASN, and (**I**) carnitine palmitoyl transferase (CPT)1 mRNA levels. in the interscapular BAT of control rats (C), pair-fed rats (PF) and rats receiving a chronic icv leptin infusion (L). Data are presented as means ± SEM. N = 5. DU, densitometry units; NS, non-significant. * *p* < 0.05 vs. C; ^#^
*p* < 0.05 vs. PF.

**Figure 6 ijms-22-02827-f006:**
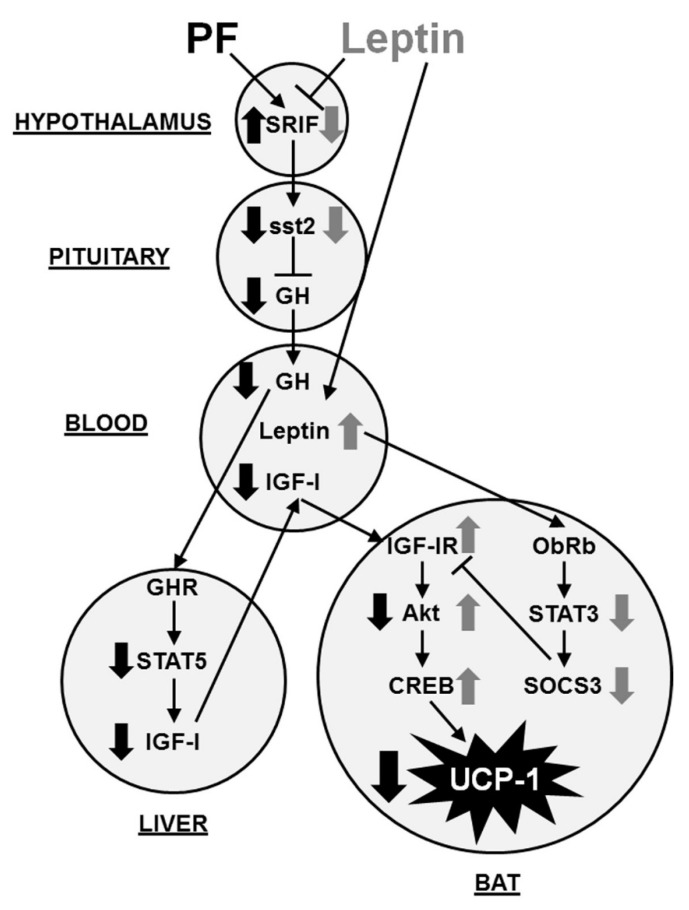
Model for modulation of the response of brown adipose tissue by leptin in a situation of negative energy balance. Pair-feeding increases SRIF and leptin infusion reduces it. SRIF inhibits GH synthesis and secretion, diminishing serum GH levels only in pair-fed rats. Thus, hepatic STAT5 activation is reduced and serum IGF-I levels are decreased in pair-fed rats. Chronic central leptin infusion increases serum leptin levels and reduces STAT3 activation, as well as SOCS3 levels in BAT. The association of SOCS3 to IGF-IR is decreased in leptin-treated rats, increasing IGF-IR activation and subsequent intracellular signaling. Thus, Akt and CREB activation is higher in leptin-treated rats, inhibiting the reduction in UCP-1 expression associated to a negative energy balance. Thin arrows indicate activation and blocked lines indicae inhibition. Thick black arrows denote pair-fed rats and thick grey arrows indicate leptin-treated rats. Akt, protein kinase B; BAT, brown adipose tissue; CREB, cAMP response element binding protein; GH, growth hormone; GHR, GH receptor; IGF-I, insulin-like growth factor-I; IGF-IR, IGF-I receptor; ObRb, long form of the leptin receptor; PF, pair-fed; SOCS3, suppressor of cytokine signaling 3; SRIF, somatostatin; sst2, somatostatin receptor 2; STAT, signal transducer and activator of transcription; UCP-1, uncoupling protein-1.

**Table 1 ijms-22-02827-t001:** Glycemia, insulinemia, and signaling parameters and weight of interscapular brown adipose tissue.

	Control	Pair-Fed	Leptin
Free IGF-I (ng/mg protein)	ND	ND	0.17 ± 0.02
Glucose (mg/dL)	78.3 ± 4.2	75.0 ± 5.6	84.1 ± 4.9
Insulin (ng/mL)	0.68 ± 0.12	0.70 ± 0.13	0.89 ± 0.24
IRβ	100.0 ± 24.8	83.0 ± 5.6	74.8 ± 10.5
PPARγ	100.0 ± 10.4	111.9 ± 16.5	102.7 ± 9.1
PTP1B	100.0 ± 13.2	90.0 ± 5.5	87.3 ± 3.9
pTyr694-STAT5	100.0 ± 14.7	81.2 ± 11.5	106.0 ± 12.6
Weight (g)	0.69 ± 0.08	0.45 ± 0.08 *	0.56 ± 0.10

Value are means ± SEM of five animals. BAT, interscapular brown adipose tissue; IRβ, beta chain of the insulin receptor; ND, not detectable; p, phosphorylated; PPARγ, peroxisome proliferator-activated receptor-γ; PTP1B, protein tyrosine phosphatase 1B; pTyr694-STAT5, of protein tyrosine phosphatase 1B phosphorylated at Tyr694. * *p* < 0.05 vs. C.

## Data Availability

All relevant data are included within the manuscript.

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
