# Peer review of "Leptin Modulates the Response of Brown Adipose Tissue to Negative Energy Balance: Implication of the GH/IGF-I Axis"

_ijms, 2021, doi:10.3390/ijms22062827_

Round 1
Reviewer 1 Report
The manuscript is well done
Author Response
Reviewer 1
Comments and Suggestions for Authors
The manuscript is well done
Response: The authors wish to thank the reviewer for their comment regarding our manuscript.
Reviewer 2 Report
Title:
Leptin modulates the response of brown adipose tissue to negative energy balance: Implication of the GH/IGF-I axis.
Authors:
Vicente Barrios, Laura M. Frago, Sandra Canelles, Santiago Guerra-Cantera, Eduardo Arilla- Ferreiro, Julie A. Chowen and Jesús Argente.
General Comment
Leptin affects energy balance by regulation of food intake and energy expenditure. Whereas there is a plethora of data on the role of leptin in appetite control, its role in thermoregulation and brown adipose tissue (BAT) activity is much less known (as suggested in Endocr Rev. 2020 Apr 1;41(2):232–60). In their study, Barrios et al. investigated effects of chronic central leptin infusion with pair-fed rats on the GH/IGF-I axis and BAT activity and metabolism. They found that central leptin infusion due to activation of IGF-I signaling and the improvement of the inflammatory environment in BAT can maintain thermogenesis despite caloric restriction. The study is well designed to verify the research hypothesis, and the results constitute an essential contribution to the present state of knowledge. Therefore I have only minor comments I would like the Authors to consider before the manuscript is accepted for publication.
Minor revisions:
- please consider summarizing the main study findings in a figure (as the Authors did previously in Mol Cell Endocrinol. 2015 Nov 5;415:157-72. )
- please explain the abbreviations when they appear in the text for the first time (namely, since the material and methods section is at the end of the manuscript, the abbreviations should be explained in the result section).
- please consider the assistance of the native speaker in order to polish some linguistic stumbles within the manuscript, e.g.:
“In this way, although both groups have a reduction of food intake, leptin-infused group have an excess of hypothalamic leptin, which would indicate to the neuronal circuits controlling metabolic homeostasis a positive energy balance” – please correct the subject/verb disagreement.
“Thus, together these data suggest that the reduction in inflammatory markers and its signaling in leptin-treated rats may link activation of IGF-I signaling and thermogenesis."- if "its" refers to "inflammatory markers," consider changing to "their."
“In summary, our results suggest that leptin infusion maintains thermogenesis in a situation of caloric restriction. Changes in lipid anabolism play a key role (consider adding in this phenomenon? process?), and this could be due to activation of IGF-I signaling, that may be associated with the reduction in the association of SOCS3 to the IGF-IR and the improvement of the inflammatory environment in BAT as well as to the preservation of GH-IGF-I axis”.
Author Response
Reviewer 2
General Comment
Leptin affects energy balance by regulation of food intake and energy expenditure. Whereas there is a plethora of data on the role of leptin in appetite control, its role in thermoregulation and brown adipose tissue (BAT) activity is much less known (as suggested in Endocr Rev. 2020 Apr 1;41(2):232–60). In their study, Barrios et al. investigated effects of chronic central leptin infusion with pair-fed rats on the GH/IGF-I axis and BAT activity and metabolism. They found that central leptin infusion due to activation of IGF-I signaling and the improvement of the inflammatory environment in BAT can maintain thermogenesis despite caloric restriction. The study is well designed to verify the research hypothesis, and the results constitute an essential contribution to the present state of knowledge. Therefore I have only minor comments I would like the Authors to consider before the manuscript is accepted for publication.
Minor revisions:
- please consider summarizing the main study findings in a figure (as the Authors did previously in Mol Cell Endocrinol. 2015 Nov 5;415:157-72. )
Response: We would like to thank the Reviewer for his/her constructive observations. We have included a diagram summarizing the main study findings (Figure 6, page 12).
- please explain the abbreviations when they appear in the text for the first time (namely, since the material and methods section is at the end of the manuscript, the abbreviations should be explained in the result section).
Response: We have explained the abbreviations when they appear for the first time (usually, Introduction and Results sections).
- please consider the assistance of the native speaker in order to polish some linguistic stumbles within the manuscript, e.g.:
“In this way, although both groups have a reduction of food intake, leptin-infused group have an excess of hypothalamic leptin, which would indicate to the neuronal circuits controlling metabolic homeostasis a positive energy balance” – please correct the subject/verb disagreement.
Response: We have amended the verb disagreement (third paragraph of the Discussion section, page 7).
“Thus, together these data suggest that the reduction in inflammatory markers and its signaling in leptin-treated rats may link activation of IGF-I signaling and thermogenesis."- if "its" refers to "inflammatory markers," consider changing to "their."
Response: We have changed “its” by “their” because we are referring to inflammatory markers (eighth paragraph of Discussion, page 8).
“In summary, our results suggest that leptin infusion maintains thermogenesis in a situation of caloric restriction. Changes in lipid anabolism play a key role (consider adding in this phenomenon? process?), and this could be due to activation of IGF-I signaling, that may be associated with the reduction in the association of SOCS3 to the IGF-IR and the improvement of the inflammatory environment in BAT as well as to the preservation of GH-IGF-I axis”.
Response: We have included the term process in this sentence in Conclusions (page 11).
Reviewer 3 Report
This study provides important piece of information about role of Leptin in maintaining the energy balance in brown adipose. However, it is not convincing enough to come to definite conclusion. Authors have done variety of experiments using numerous targets involved in the pathways, that made the whole reading experience extremely confusing. Focused approach directed towards the relevance of GH/IGF is missing, which is the main theme of this paper. The work need more attention for readability and assimilation of the available information into a digestible form.
- The authors should quantify the amount of protein expressed in addition to mRNA content to prove that target proteins are translated.
- Experimental methodologies should be explained properly. For e.g. what are pair-fed mice and how are they used for experiments.
- Few sentences need rearrangement to convey message accurately. for e.g. line 75 - 80, line 89 etc
- Why is the amount of GH more in the leptin infused mice (Fig 1D) when the mRNA content is less than the control in Fig 1C.
- Very few experiments are done which establish direct role of GH/IGF-I in the modulation of thermogenesis.
I do not recommend this work for publication in the present form.
Author Response
Reviewer 3
This study provides important piece of information about role of Leptin in maintaining the energy balance in brown adipose. However, it is not convincing enough to come to definite conclusion. Authors have done variety of experiments using numerous targets involved in the pathways, that made the whole reading experience extremely confusing. Focused approach directed towards the relevance of GH/IGF is missing, which is the main theme of this paper. The work need more attention for readability and assimilation of the available information into a digestible form.
1.The authors should quantify the amount of protein expressed in addition to mRNA content to prove that target proteins are translated.
Response: The authors thank the Reviewer for his/her constructive comments. The concentration of most of the proteins are included in this version of the manuscript. The results are show in subsection 2.2. “Effects of food deprivation and leptin infusion on the GH/IGF-I axis”, page 2 and in Fig. 1D, 1E and 1F, page 3 and in subsection 2.6. “Changes in variables involved in lipid metabolism” page 5 and 6 and in Fig. 5F and 5H, page 6.
2.Experimental methodologies should be explained properly. For e.g. what are pair-fed mice and how are they used for experiments.
Response: We have explained the meaning of the pair-fed group and the reason to be included in the last paragraph of the Introduction section (page 2) and in subsection 4.3. (page 9).
In addition, we have described the new ELISA for determination of protein levels of somatostatin in the hypothalamus (subsection 4.5.3., page 10) and new antibodies and the normalization of gel loading variabilities for the quantification of new proteins suggested by the Reviewer.
3.Few sentences need rearrangement to convey message accurately. for e.g. line 75 - 80, line 89 etc
Response: We have restructured these sentences in subsections 2.1 and 2.2. (page 2).
4.Why is the amount of GH more in the leptin infused mice (Fig 1D) when the mRNA content is less than the control in Fig 1C.
Response: Although serum GH levels tend to be higher in leptin-treated rats, there are no significant differences with respect to the control group. This upward tendency may be due to a greater GH secretion, as suggested by the reduced GH protein concentration in the pituitary of leptin-treated rats with respect to controls. These data are in agreement with the diminished SRIF mRNA levels in the leptin group, since SRIF inhibits not only the synthesis, but also (and mainly) the secretion of GH.
This interesting aspect suggested by the Reviewer has been included in Discussion (second paragraph, page 7).
5.Very few experiments are done which establish direct role of GH/IGF-I in the modulation of thermogenesis.
Response: We agree with the reviewer that additional experiments would be needed to establish a direct association between GH/IGF-I and modulation of thermogenesis. This caveat has now been included in the Discussion section (second to last paragraph of Discussion, page 9). In spite of this limitation, our data indicate an increase in GH expression and serum GH levels, as well as the increase not only in serum IGF-I, but also in the brown adipose tissue. This is in concordance with the activation of IGF-I receptor, Akt and CREB in the absence of changes in the phosphorylation of insulin receptor.
I do not recommend this work for publication in the present form.
Round 2
Reviewer 3 Report
I am satisfied with the answers provided by the authors and agree to accept the manuscript.